# The Graphite Occurrences of Northern Norway, a Review of Geology, Geophysics, and Resources

**Håvard Gautneb [1],\*, Jan Steinar Rønning [1,2], Ane K. Engvik [1], Iain H.C. Henderson [1], Bjørn Eskil Larsen [1], Janja Knežević Solberg [1], Frode Ofstad [1], Jomar Gellein [1], Harald Elvebakk [1] and Børre Davidsen [1]**.

[1] Geological Survey of Norway P.O. BOX 6315, NO 7491 Trondheim, Norway; Jan.Ronning@ngu.no (J.S.R.); Ane.Engvik@NGU.NO (A.K.E.); Iain.Henderson@NGU.NO (I.H.C.H.); Bjorn.Larsen@NGU.NO (B.E.L.); Janja.Knezevic@NGU.NO (J.K.S.); Frode.Ofstad@NGU.NO (F.O.); Jomar.Gellein@NGU.NO (J.G.); Harald.Elvebakk@NGU.NO (H.E.); Borre.Davidsen@NGU.NO (B.D.)

[2] Department of Geoscience and Petroleum, Norwegian University of Science and Technology, NO-7491 Trondheim, Norway

\* Correspondence: havard.gautneb@ngu.no Tel: +47-9092-8332

**Abstract:** There are three provinces in Northern Norway in which occurrences of graphite are abundant; the Island of Senja, the Vesterålen archipelago, and the Holandsfjorden area. From these provinces, we report graphite resources from 28 occurrences. We use a combination of airborne and ground geophysics to estimate the dimensions of the mineralized areas, and, combined with sampling and analysis of the graphite contents, this gives us inferred resources for almost all the occurrences. The average TC (total carbon) content is 11.6%, and the average size is 9.3 Mt or 0.8 Mt of contained graphite. We demonstrate that the Norwegian graphite occurrences have grades and tonnages of the same order of magnitude as reported elsewhere. The graphite-bearing rocks occur in a sequence that encompasses carbonates, meta-arenites, acid to intermediate pyroxene gneisses, and banded iron formations metamorphosed into the granulite facies. Available radiometric dating shows that the graphite-bearing rocks are predated by Archean gneisses and postdated by Proterozoic intrusions of granitic to intermediate compositions.

**Keywords:** flake graphite; critical minerals; mineral resources; geophysics; Norway

## 1. Introduction

Norway has been a producer of graphite for more than 100 years and is currently one of the major suppliers of natural graphite to Europe from a European source. Following the European and American methodology for assessment of criticality, graphite is a critical mineral [1,2]. Graphite is also an essential mineral in "green" technology and is one of the most essential components of Li-ion batteries, which will be vital in a sustainable green future. The worldwide demand for graphite is expected to show considerable growth in the near future [3].

Graphite, when compared to most other industrial minerals, is an expensive mineral product: the price strongly depends, however, on two factors, one being the flake size and the other being product purity. The price difference between medium and large flake sizes is usually about 1.5× (c. $/t1000 to $/t1500). The price difference between a 95% carbon medium flake and a 99.9% carbon battery-grade flake can be up to about 20× ($/t 1000 to $/t 20.000) (see www.indmin.com).

Somewhat simplified, graphite mineralization is the result of release of volatiles such as $CH_4$, $CO_2$, and $H_2O$ from carbonaceous material during metamorphism and subsequent reduction of the carbon component and formation of graphite. If this process take place "in situ", in the rock, the result

is occurrences of flake graphite: if the volatiles migrate in hydrothermal solutions, the carbon component can be deposited in, for example, vein-type deposits [4]. Most studies on graphite mineralization describe the first or the second of these two deposit types [4–25] or investigate graphite's influence on crustal rheology and tectonics [26–34]. There are, essentially, four methods for characterizing graphite and graphite-forming processes: in addition to standard chemical analysis, these are: (1) vitrinite reflectance, e.g., [35–37], (2) X-ray diffraction, e.g., [23,38–40], (3) Raman spectroscopy, e.g., [5,25,27,29,41–45], and (4) stable isotopes, e.g., [12,18–21,24,46–49]. These methods are commonly used in combination. Following the recommendations of [50], we regard a rock as graphite-bearing if its total carbon (TC) is >0.5%.

Graphite is a common mineral in metamorphic rocks in Norway. However, localities with enrichments to levels that are regarded as economically interesting are limited to a few provinces. There are, in Norway, four known provinces in which graphite mineralizations are found and commercial mining of graphite has taken place (Figure 1) [51,52]. From north to south these provinces are: (1) the island of Senja, (Figure 2a), (2) the Vesterålen islands, (Figure 2b), (3) the Holandsfjorden area, (Figure 2c), and (4) the Bamble area in southern Norway. The graphite occurrences in the Bamble area were described by [53,54] and will not be discussed here. All the graphite occurrences on the Fennoscandian shield occur in Proterozoic rocks of high metamorphic grade. Graphite is abundant within such areas at several localities in northern Norway. The most famous is the Skaland graphite mine, where the world's richest flake graphite schist is being mined [55].

There are, with some exceptions [56–58], quite few published studies on the geology and geophysics of graphite ore deposits from the Fennoscandian Shield and elsewhere, compared to those on other commodities. We aim, in this paper, to present an overview of the geological setting, field relationships, and similarities between the different graphite provinces in North Norway. Our descriptions will be focused on localities where we believe that there is or can be an economical potential. To keep this review paper as concise as possible, we limit our descriptions to key occurrences, where the field relationships, approach, and methods can be easily illustrated. This will then be the basis for our overview and discussions. In addition, we will review methods used and results from bench-scale ore dressing experiments based on our materials and will also compare our resource estimates with results reported from elsewhere. We will also limit our descriptions of mineral chemistry, isotopes, structural, and tectono-metamorphic processes to what is necessary for our discussions. Detailed discussions on these topics are in preparation and will be reported elsewhere. We also regard modeling of predictive ore exploration as being beyond the scope of this study.

This review is based on material that has been collected over a period of 70 years, some of which is unpublished and accessible only in Norwegian. The bulk of work has been carried out after 1990, especially after 2013. The different investigations, within such a long period, differ in approach, aim, and methods, and the available data differ therefore considerably between the described areas. We have, however, deliberately included an account of previous investigations and referred to older work that we consider to be relevant and thus make these known to others who want to pursue graphite investigations in Norway. Our main goal for the graphite projects has been to find economically viable occurrences, using all the available methods used in mineral exploration, field geology, geophysics, geochemistry, and ore-dressing tests.

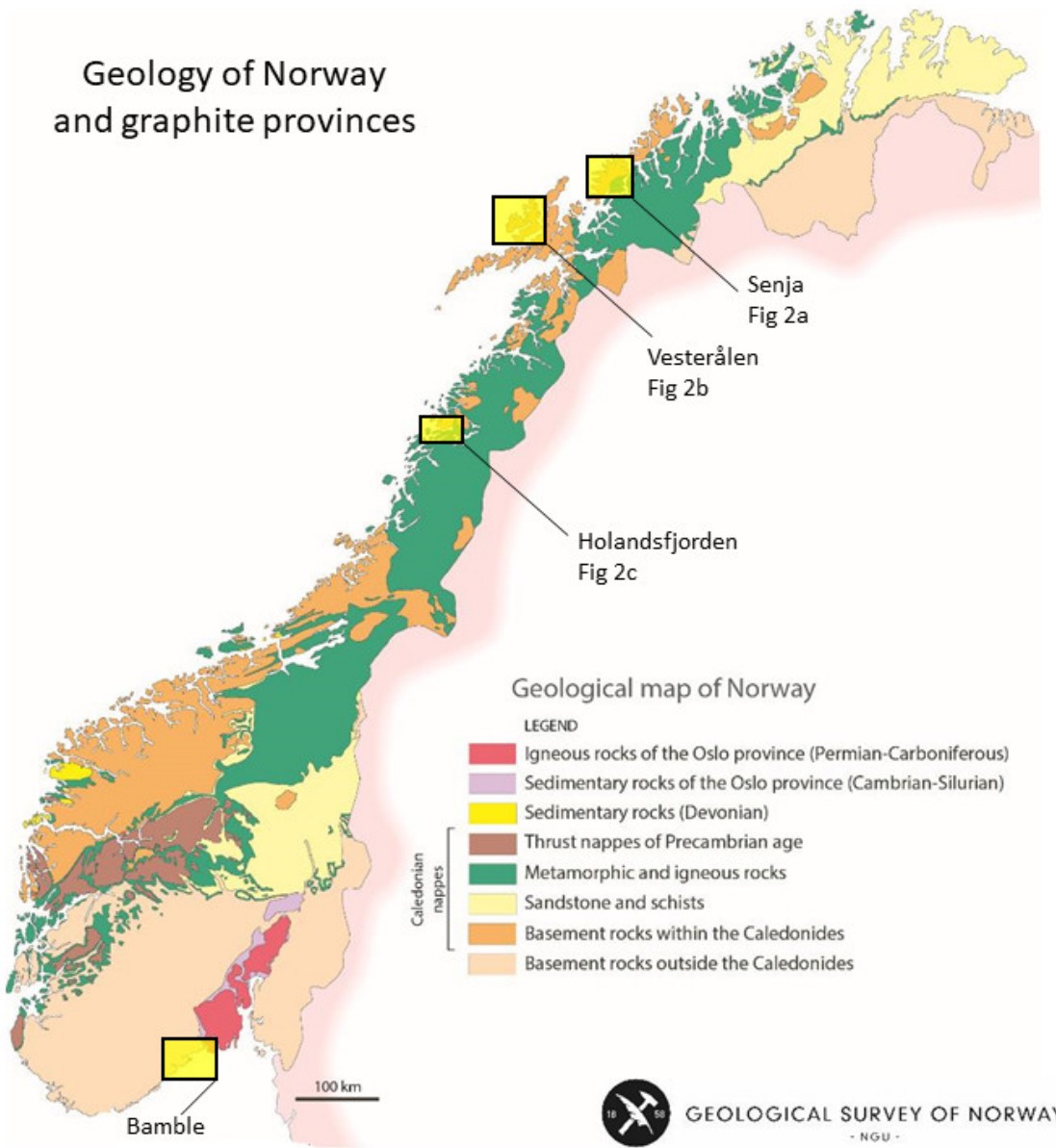

**Figure 1.** Geology of Norway and the location of graphite provinces.

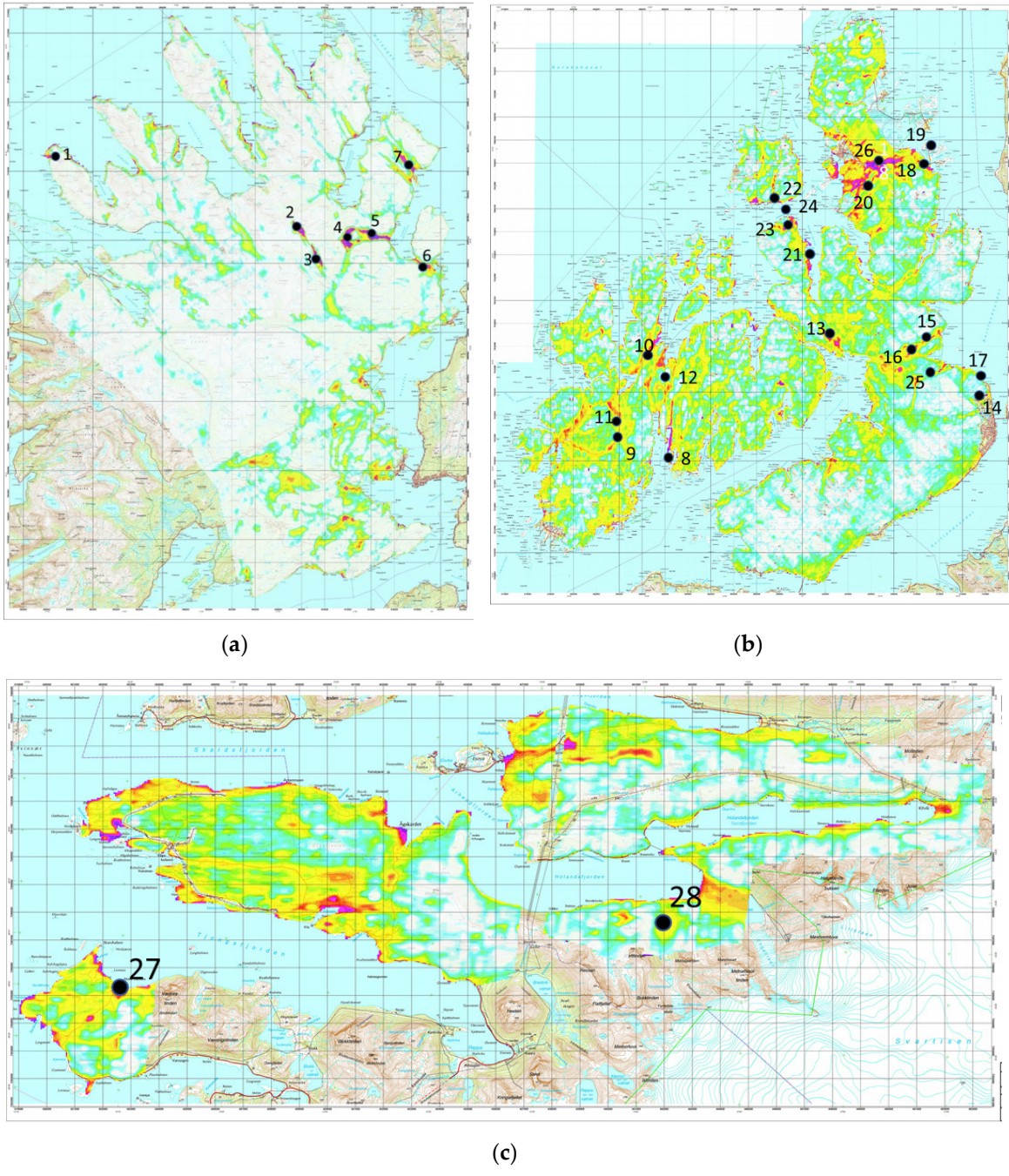

**Figure 2.** Maps of airborne EM (7000 Hz) with graphite occurrences: (**a**) Senja, (**b**) Vesterålen, and (**c**) Holandsfjorden. The names of occurrences are listed in Table 1, compiled from data reported in [51,59,60]. See links in the text for access to map files in high resolution.

**Table 1.** Investigated graphite occurrences, their average total carbon (TC) content (Wt. %), estimated tonnage, and calculated contained graphite. See the text for the approach used for calculating tonnages and contained graphite contents.

| Occurrence No. | Occurrence Name | % TC (mean) | Tonnage (Mt) | Contained Graphite (Mt) | References |
|---|---|---|---|---|---|
| 1 | Trælen | 31.0 | 1.80 | 0.56 | * |
| 2 | Vardfjellet | 9.2 | 12.84 | 1.18 | [51,61] |
| 3 | Hesten | 5.8 | 2.07 | 0.12 | [51,61] |
| 4 | Bukken | 6.5 | 51.03 | 3.34 | [51,61,62] |
| 5 | Litljkollen | 5.3 | 34.54 | 1.83 | [51,61] |
| 6 | Grunnvåg | 5.2 | 22.77 | 1.19 | [51,61] |
| 7 | Skardsvåg | 2.1 | n. a. | n. a. | [51] |
| 8 | Haugsnes | 16.2 | 8.40 | 1.36 | [52,60,63] |
| 9 | Kjerkhaugen | 6.5 | 2.42 | 0.16 | [60] |
| 10 | Møkland | 13.2 | 3.40 | 0.45 | [52,60,63] |
| 11 | Rise | 7.9 | 0.19 | 0.01 | [60] |
| 12 | Sommarland | 12.5 | 0.85 | 0.11 | [52,60,63] |
| 13 | Brenna | 10.1 | 7.94 | 0.80 | [60] |
| 14 | Evassåsen | 7.6 | 2.12 | 0.16 | [60] |
| 15 | Vikeid Central | 13.8 | 8.89 | 1.23 | [60] |
| 16 | Vikeid West | 11.3 | 29.63 | 3.35 | [60] |
| 17 | Ånstad | 36.8 | 0.21 | 0.08 | [60] |
| 18 | Alsvåg | 8.9 | 0.25 | 0.02 | [60] |
| 19 | Instøya | 9.3 | 14.82 | 1.42 | [60] |
| 20 | Rødhamran | 14.8 | 1.38 | 0.20 | [52,60,63] |
| 21 | Romset | 14.7 | 9.63 | 1.42 | [60] |
| 22 | Skogsøya | 20.0 | 1.42 | 0.30 | [52,60,63] |
| 23 | Smines | 7.1 | 18.89 | 1.34 | [52,60,63] |
| 24 | Svinøya | 11.7 | 0.02 | 0.02 | [52,60,63] |
| 25 | Jennestad | 9.6 | 3.66 | 0.35 | [56,58,64] |
| 26 | Myre | na | na | na | [52,60,63] |
| 27 | Nord-Værnes | 4.1 | 0.60 | 0.02 | [59] |
| 28 | Rendalsvik | 11.1 | 1.90 | 0.21 | [65,66] |
| | Average | 11.6 | 9.3 | 0.81 | |
| | Sum | | 241.6 | 21.51 | |

* Pers. comm., T. Abelsen, Skaland graphite.

## 2. Geological Setting

Graphite is a common mineral in meta-supracrustal rocks of mid- to late-Proterozoic age on the Fennoscandian shield. There are more than 50 occurrences in Finland, Sweden, and Norway [55,57,67]. The graphite occurrences from Senja, Vesterålen, and Holandsfjord areas are all located within Archaean to Proterozoic basement provinces that occur as basement windows within younger Caledonian rocks.

*2.1. Senja*

The island of Senja is located in the southern part of the West Troms Basement Complex (WTBC), which is an Archean to Paleoproterozoic basement horst west of the Caledonian orogenic rocks. The WTBC comprises a unit of 2.89–2.70 Ga tonalitic and granitoid gneisses that is intruded by a mafic dyke swarm. The basement gneisses are overlain by numerous supracrustal units that comprise metapsammites, conglomerates, metavolcanites, dolomitic carbonates, banded iron formations, graphite schists, and massive sulphide deposits [68,69] of varying ages [70]. These basement gneisses

and supracrustal cover rocks were deformed during the Svecofennian orogeny at 1.8–1.75 Ga, and peak metamorphism varies within the complex from low grade in the north to amphibolite to granulite facies in the southern area (Senja). A suite of bimodal plutonism occurred at 1.8–1.79 Ga, synchronous with a major suite of plutonic rocks in Vesterålen and Lofoten.

## 2.2. Vesterålen Area

The rocks of the Lofoten–Vesterålen archipelago are regarded as the southern continuation of the WTBC [71,72]. They form of an Archean to Proterozoic basement consisting of both granitic gneisses and migmatites and gneisses of supracrustal origin, in which zircon dates range from 2.7 Ga to 2.6 Ga. A sequence of supracrustal rocks, comprising of gneisses and amphibolites of intermediate to mafic composition is believed to represent metavolcanics, metapsammites, dolomite marbles, graphite schist, and banded iron formation [73]. These supracrustal rocks are very heterogeneous and at most places they are orthopyroxene gneisses of variable compositions. Granulite-facies peak metamorphic conditions have been estimated to be 810–835 °C at 0.73–0.77 GPa [74]. These units were intruded at about 1800–1790 Ma [71] by an anorthosite–mangerite charnockite–granite (AMCG) suite of rocks that intruded at 0.4 GPa and 800–925 °C [75].

## 2.3. Holandsfjord Area

The Holandsfjorden area comprises rocks similar to those in the Senja and Lofoten–Vesterålen areas. A unit of paragneisses with different garnet-mica gneisses includes, locally, bands of graphite schist, and is intruded by different types of porphyritic or equigranular granites [76]. U/Pb dates of the granitic rocks range in age from about 1.88 Ga [77], thus about similar ages as the AMCG of the Lofoten–Vesterålen area.

There are, in the Holandsfjorden area, two localities where graphite has been mined—the Rendalsvik and the Nord-Værnes deposits. Both were in production from 1933 to 1945 [78,79]. The geology adjacent to the older mines was described by [65,80] for the Rendalsvik area and by [59] for Nord-Værnes.

## 3. Previous and Recent Work

Graphite occurrences in Norway were first described by [81,82]. The history of mining and development of the graphite industry has been described by [83,84]. In the 1950s, most graphite occurrences in Northern Norway were investigated for their possible co-occurrence of uranium, which turned out to be negative for all the occurrences [85].

The Skaland graphite deposit was put into periodic production in 1922 and, from 1932 to the present, the mine has been in continuous production. From 2006, the mine location has been at Trælen [86]. The Trælen graphite mine (1 on Figure 2a) is the world's richest flake graphite mine in operation, with a TC (total carbon) content of about 30% [55]. The geology of the Skaland mine is described by [87]. The geophysical features of the Skaland/Trælen and some other occurrences is described by [62,88–90]. The structural geology of most of the graphite occurrences on Senja was described by [91]. New airborne geophysical data were reported by [92]. Raman spectroscopy of graphite from Skaland is reported by [25], who show that the graphite is a fully ordered flake graphite.

Graphite was mined at several localities in Vesterålen from about 1890 to 1918, and again from 1949 to 1960. During the latter period, the geology and geophysics of the graphite deposits was described in several unpublished reports [93–96]. About 770 m of underground adits and drifts as well as several surface trenches were made during the 1950s. The general geology was described by [97]. Detailed airborne geophysical data, including magnetic, radiometric, and electromagnetic (EM) data, were collected in two periods (1988 and 2012) and are reported in [98–100]. Follow-up work on the ground, including geophysics, trenching, and drilling has been described in reports in Norwegian [101–108] and this work was reviewed by [58]. The junior mining company *Norwegian graphite* reinvestigated the graphite deposits in the Jennestad area in 2013 (25 on Figure 2b). Their work involved drilling of 13 drill holes with an aggregate length of 1360 m. Indicated resources were

estimated to be 3.66 Mt with 9.6% total graphite [64]. *Norwegian graphite* was liquidated in 2015 and their data have been released to the Geological Survey of Norway (NGU), (see supplementary data). Raman spectroscopy of Jennestad graphite was reported by [56] and showed that the graphite is fully ordered and almost free of internal defects.

The Holandsfjorden with the Nord-Værnes (27 on Figure 2c) and the Rendalsvik (28 on Figure 2c) abandoned graphite mines were mapped by [65,80]. Ground geophysics (turam) was reported by NGU [109]. The occurrence was investigated with respect to the co-occurrences of uranium [110]. The historical mine production (1933–1945) was 8170 tons [78]. The endowment was measured by [66], who indicated resources of 2.3 Mt with a mean TC of 11.1%. New airborne and ground geophysics together with geology of the graphite occurrences was reported by [59,111].

## 4. Materials and Methods

### 4.1. Geophysical Methods

Graphite schists are, on the Fennoscandian shield, usually the least exposed rocks in a given area. Even if the overburden is thin, geophysical methods are essential for locating and estimating the size of individual occurrences. Graphite has a clear geophysical signature due to its good electric conductivity, and electromagnetic and electric methods are well suited and essential for the location and characterization of the graphite mineralization that are covered.

Airborne geophysical surveys have been carried out in all the three graphite provinces in 2012, 2013, and 2014 including measurements using magnetic (MAG), five-frequency electromagnetic (EM), and radiometric (Gamma spectrometry, RAD) instrumentation. Details regarding the instrumentation and resulting datasets are described in [90,92,99,111]. The apparent electric conductivity of the ground, calculated for each frequency using a half space model (Oasis montaj v.9.7, Geosoft Inc., Toronto, ON, Canada), is particularly used for finding graphite mineralization. Airborne geophysical maps with all the occurrences reported in this paper are shown in Figure 2a–c and listed in Table 1. Some of the different anomalies found on the airborne geophysical data were known to be associated with graphite from previous investigations, but their areal extent became better defined in the later studies. In addition, numerous new anomalies were located, and almost all the anomalies were then targets for ground geophysical and geological follow up work. Helicopter-borne geophysical data are downloadable from https://www.ngu.no/en/page/other-maps-and-data.

The follow-up on the ground, at almost every target, included the following geophysical methods, (see [51,60] for details):

- Ground EM measurements with the use of EM31 [112].
- Charged- and self-potential (CP/SP), using in-house-developed equipment.
- 2D electric resistivity traversing (ERT) and induced polarization (IP), using the Lund system [113] and multigradient electrode configuration. The measurement was done with an ABEM Terrameter LS [114], and data inversion was carried out using the software RES2DInv [115].

### 4.2. Petrophysical Measurements

Petrophysical parameters including volume, density, pore-volume, porosity, susceptibility, remanence, heat-conductivity, specific heat capacity, together with data on total sulphur and total carbon on 125 samples are included in the supplementary data. The average density is 2437 kg/t.: this value is used in the resource estimates. Because many of the samples have a high graphite content and are thus electrically conducting, the measurements of susceptibility must be used with care.

### 4.3. Geological Methods

The graphite-bearing units were sampled at exposures, in trenches, or from drill cores. Basic geological and structural data were collected concurrently. Detailed petrographic studies were performed by optical microscopy and scanning electron microscopy (SEM) using a LEO1450 VP instrument at the Geological Survey of Norway (NGU, Trondheim, Norway). Modal contents of

graphite as well as flake grain sizes were measured by image processing using the ZEN blue™ software (version 2.5) from Zeiss (Köln, Germany).

### 4.4. Geochemical Methods

The graphite content of the samples was analyzed using a LECO SC 632 carbon and sulphur analyzer (LECO, St. Joseph, MI, USA). Three variants of LECO type procedures are normally used for graphite analysis. After crushing and milling the graphite, the content can be determined by: (A) removing any inorganic carbonates by acid, roasting the sample to remove organic plant material, and finally analysis using a LECO type of instrument; (B) analysis of total organic carbon (TOC) similar to A, without the roasting step; and (C) measurement only by Leco analyzer (, without removing carbonates or organic matter: the latter method gives the total carbon (TC). Rønning et al. [60] reports both TOC and TC for a number of samples and compared these methods and showed that for the majority of our samples, TOC and TC only deviated within the analytical uncertainty. Therefore, on >90% of our samples only method C was used, which is the fastest and least costly. Table 1 shows the average % TC for our occurrences, and the complete dataset of 679 analysis of TC, TOC, and total sulphur (TS) analyses are reported in [51,60], data used in this paper is available in the supplementary data.

### 4.5. Bench Scale Beneficiation

Two sample sets of graphite schists were tested in bench-scale graphite beneficiation [52,116]. The sample tested by [116] was from the Jennestad (deposit no 25 Table 1, Figure 2b). The samples tested by [52] were from the Møkland deposit (no 10 Table 1, Figure 2b). Both trials follow approximately the same flow sheet. Following crushing and primary grinding, a series of stepwise flotation and regrinding steps were done. The trials were run using a rod mill and a Denver laboratory flotation cell. Øzmerih [116] used MICB (methyl isobutyl carbinol) and FlotolB™ as frother, and kerosene as activator. In [52] Dowfroth, 400E™ was used as frother. See [52,116] for additional technical details.

## 5. Results

### 5.1. Geology of the Graphite-Bearing Units

The graphite schist represents the most weathered and poorly exposed rock in all three areas. Field exposures are only available on a few coastal exposures, road cuttings, and exposed flat-lying areas in the mountains. The graphite schist is typically rusty brown, strongly schistose, and commonly tightly folded (Figure 3b,c, deposits No. 4 and 6 on Figure 2a). The graphite bands can vary in thickness from cm scale to several tens of meters (Figure 3a, No. 24 on Figure 2b), and exposures can be followed for up to about 100 m (Figure 3d, No. 4 on Figure 2a). The graphite schist is usually, at well-exposed localities, part of a rock unit that includes amphibolite-bearing schists, and which is typically also intruded by different types of younger intrusion (Figure 3e). In all the described graphite provinces, graphite schist seems to be part of a succession that comprises pyroxene gneisses, quartz-feldspar-rich rocks (metapelites to meta-arenites), carbonates (mostly dolomites), and less common iron formations [52,59,64]. At localities where drill core data are available (sections over lengths up to 140 m), the graphite bearing sections appear randomly distributed with thicknesses varying from a few centimeters to ca. 10 m. The proportions of the host rock; pyroxene gneisses or quartz-feldspar rocks vary widely. In some localities, graphite schist shows a gradual transition into carbonates (Figure 4) (See supplement data for details). We believe that the relationship between graphite-rich units in succession is that they, in all probability, show that the graphite was part of a sedimentary sequence that underwent metamorphism that resulted in formation of graphite.

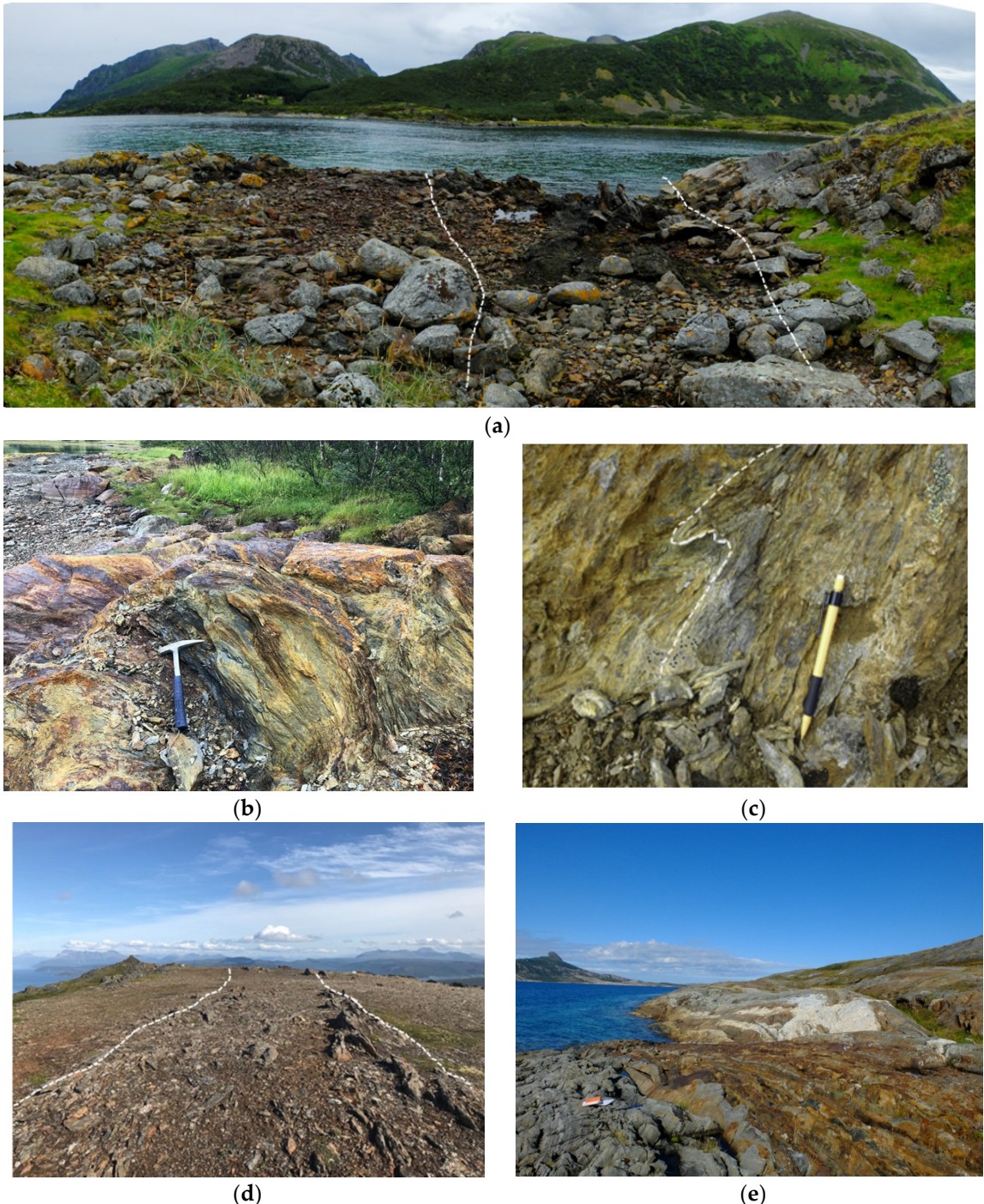

**Figure 3.** (**a**) Well-exposed massive graphite schist (Svinøya, occurrence 24). (**b**) Typical rusty, weathered, and strongly schistose graphite schist (Grunnvåg, occurrence 6). (**c**) Close-up of typically foliated and tightly folded graphite schist, foliation indicated (Vardfjellet, occurrence 2). (**d**) Outcropping graphite schist continuing about 100 m along strike (Bukken, occurrence 4). (**e**) Outcrop showing from left to right a gradual transition from amphibolitic schist into rusty graphite schist that are intruded by younger granites to the right (Nord-Værnes, occurrence 27).

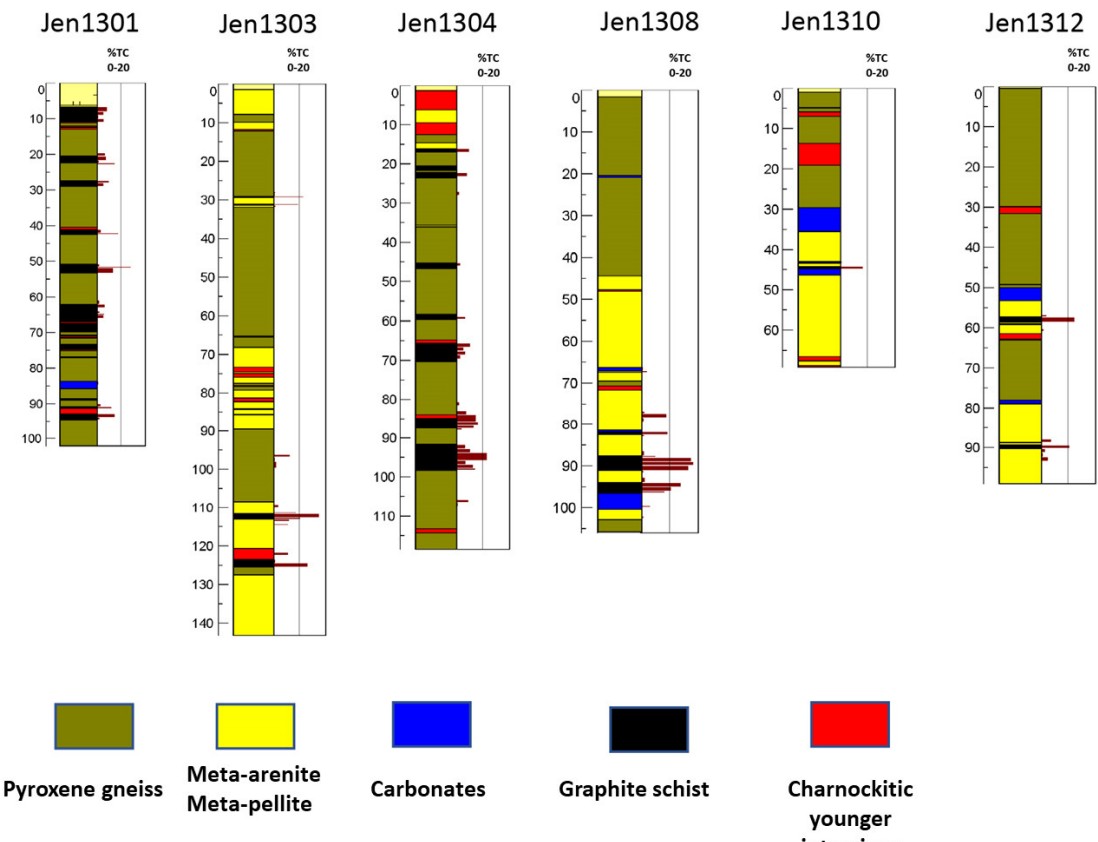

**Figure 4.** Selected drill logs based on data from Norwegian graphite (near Jennestad, occurrence 25), which show the variation and distribution of pyroxene gneisses, meta-arenite, carbonates, and graphite schist in about 100 m continuous sections together with variation in TC (compiled and simplified from Norwegian graphite, see Supplement data for details).

## 5.2. Petrography of Graphite Schist

The graphite schist has a variable grain-size, up to coarse-grained (Figure 5a). Major quantities of graphite occur together with major quantities of quartz, plagioclase, K-feldspar, and locally, of biotite, clino-, and orthopyroxenes. Ti-phases are commonly present as titanite and rutile. The graphite schist is often sulphide-rich, with pyrite, pyrrhotite, and locally chalcopyrite. Apatite is also common.

The graphite content typically ranges between 10–25 vol %, but a modal content up to 90% is found locally. Graphite crystals in hand samples usually occur as silvery shining minerals. Graphite occurs as single, well-developed flakes or as subhedral interstitial grains between silicate minerals, in aggregates 1–5 mm in size. Together with biotite, oriented graphite crystal forms a well-developed foliation and contributes to the rock schistosity. Pyroxenes, quartz, and feldspars constitute anhedral matrix minerals with metamorphic, triple-point grain boundaries and are normally fine- to medium-grained in size (Figure 5b,c).

Flake grain-size is an important parameter for the quality of a finished graphite product. In situ measurements of graphite flake grain size were reported by [58,60]. Typically, in thin section, the graphite flakes are dominated by some very few, up to ca. 3.5 mm large grains, which make up the bulk of the modal content of graphite. The mode grain size is usually much smaller, often around 300 μm. However, the in situ grain size of the rocks is of less significance. It is the grain size distribution of a finished product (after beneficiation) that matters, as discussed below.

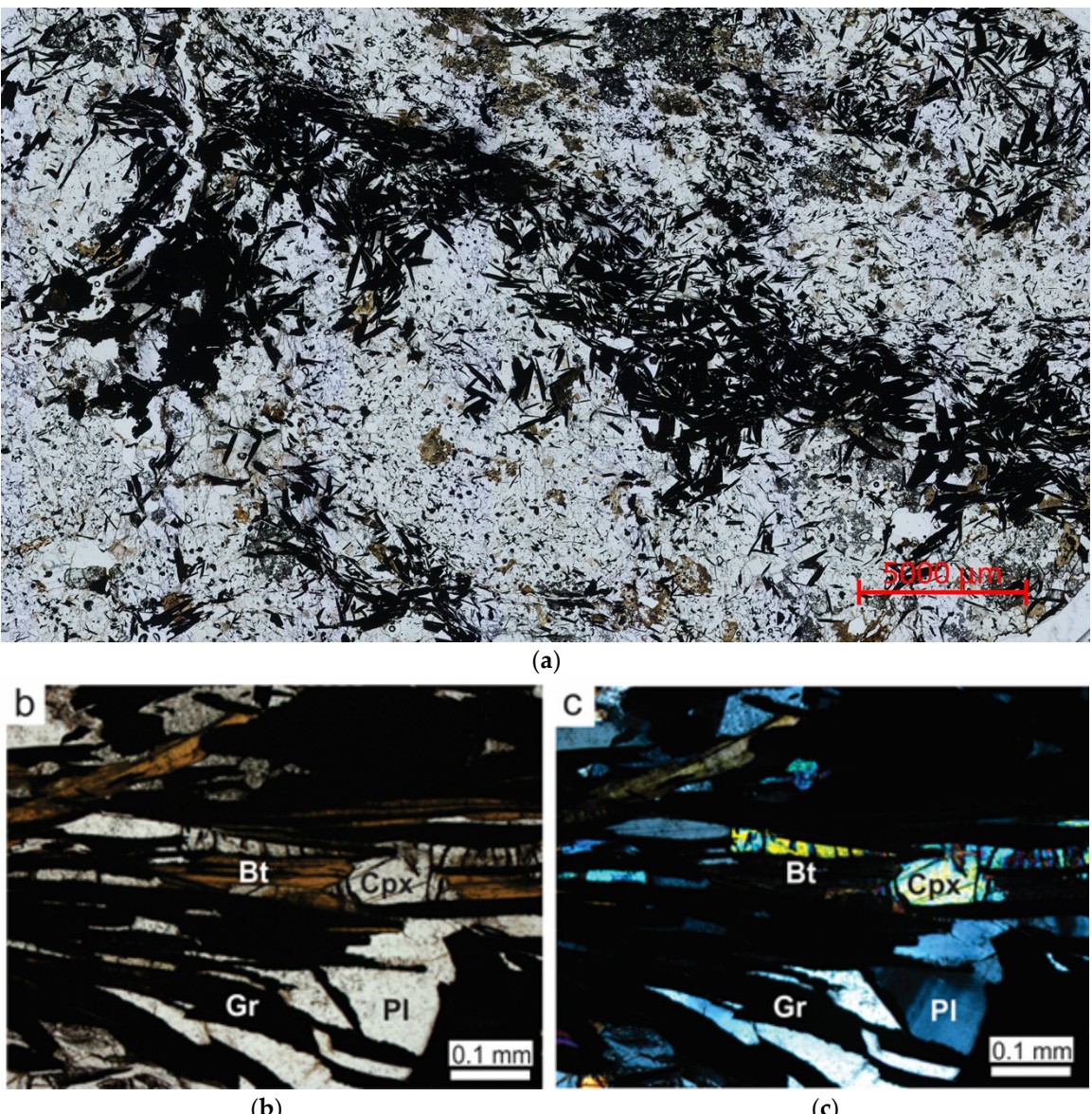

**Figure 5.** Micrographs of graphite schist: (**a**) mosaic micrograph covering a whole thin section c. 2.5 × 3.5 cm, showing the distribution crystal shape and size of the of the graphite (black flake crystals). Note the inhomogeneous nature of the aggregates of graphite crystals. Modal content is 29%. (Sample HG11-15). B–C typical graphite schist assemblage of graphite (Gr), biotite (Bt), clinopyroxene (Cpx), and plagioclase (Pl), where with foliation defined by oriented graphite crystal flakes and biotite. (**b**) Plane light and (**c**) polarized light. Sample AE171.

*5.3. Structural Geology of the Graphite-Bearing Units*

It is only on the island of Senja that there are outcrops of the graphite schist that make structural studies possible. The age relationships and the general tectonic process are described by [117]. Given the complex tectono-metamorphic history shown in the Norwegian graphite deposits, an understanding of the structural deformation is essential for any resource evaluation. We use one example to illustrate the structural complexity of the occurrences.

[118] first studied in detail the various graphite deposits on the island of Senja. [91] also characterised the complex structural episodes as having both created and modified the graphite deposits and found that the graphite is intimately associated with the development of $F_2$ folds, as graphite is best developed geographically in association with N-S-striking $F_2$ fold hinges. However, the $F_2$ fold hinges are deformed by $D_3$ deformation on approximately E-W axes. This created complex interference geometries. The graphite is most extensive geographically where $F_3$ structures intersect

graphite-bearing $F_2$ structures. The extent of graphite outcrop is also strongly affected by the presence of both $F_2$ and $F_3$ shear-zone structures, which locally ($F_2$) or regionally ($F_3$) 'shear-out' the graphite outcrops, thereby limiting the extent of the graphite deposits.

Both the $F_2$ and $F_3$ folds are highly noncylindrical in many of the deposits, demonstrating that using small-fold geometries to determine the large-scale deposit geometry is highly problematic. The complex folding and sheared nature of the graphite deposits is reflected in complex outcrop patterns. For example, Figure 6 shows the Bukken deposit (No. 4 on Figure 2a) in eastern Senja, which shows graphite tectonically thickened around $F_2$ folds and subsequently refolded by $F_3$ folds.

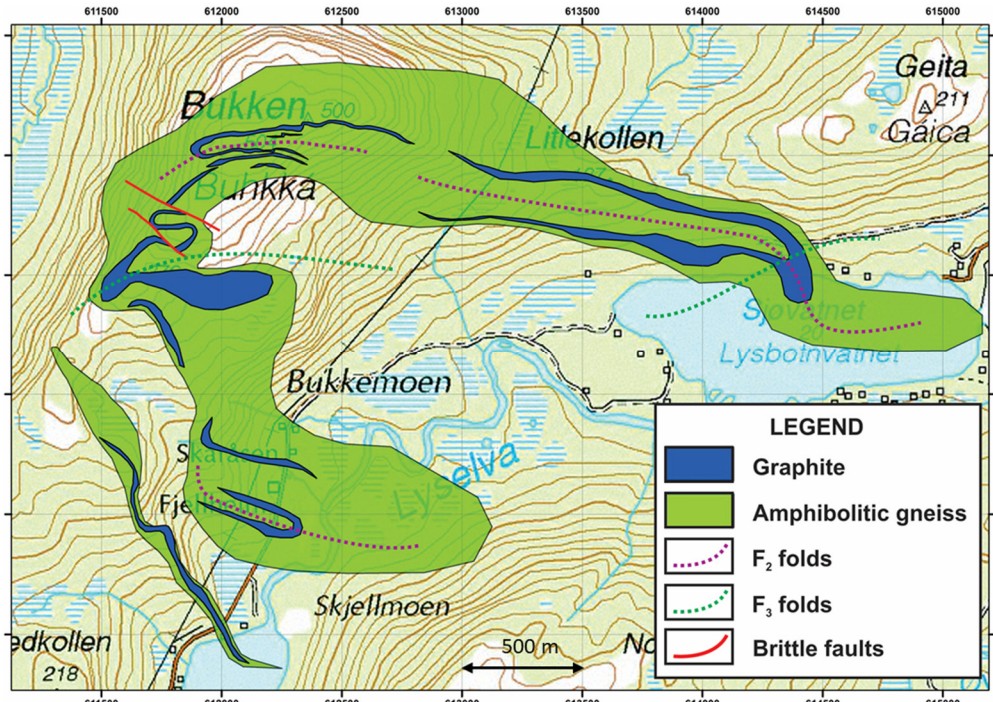

**Figure 6.** Detailed structural mapping of the Bukkemoen and Litlekollen deposits. The graphite is enveloped within amphibolitic gneiss, which forms thin lenses in the host acidic gneisses. The graphite forms thin layers up to several meters wide and is folded isoclinally around $F_2$ folds along NW-SE axes. The plunge is moderate towards the NW. The graphite is further deformed along spaced open $F_3$ fold axes with E-W trending axes, with a subhorizontal plunge. This leads to a complicated graphite outcrop pattern.

*5.4. Ground Geophysical Measurements*

The ground geophysical measurements collected at the 28 occurrences shown in Figure 2a–c had their main aim in estimating the dimensions of the mineralized bodies with respect to graphite and to measure, if possible, the number and width of the (individual) mineralized lenses. We will use two examples to illustrate our approach. In covered areas, ERT/IP were used to see if conductive structures shown in the helicopter-borne EM measurements, were caused by graphite/sulphide mineralization or by fractured/weathered rock.

The first example on Figure 7 is from Vardfjellet at Senja, (occurrence No. 2 in Figure 2a). Figure 7a shows the location of the airborne geophysical anomaly relative to neighboring anomalies. Figure 7b shows the results from EM31 apparent conductivity traversing superimposed on the airborne 7 kHz apparent resistivity data. Numerous conducting structures are seen, and, in various places, graphite schist is exposed. Sample points with TC analyses are also shown. Figure 7c shows the EM31 apparent conductivity separately and along three selected traverses where the variation in conductivity is shown to the right. Peak conductivity values are indicated with a blue arrow, which represents individual graphite schists lenses, that for the most part is overburden with a thin soil

cover. The apparent width of these lenses can then be estimated, and some peaks can be followed laterally from profile to profile. It is important to note that what apparently is a homogenous anomaly based on the helicopter data, actually comprises several subparallel, individual, highly-conductive lenses. Airborne geophysics alone would not have been able to resolve this complexity. Our combined geological and geophysical observations for Vardfjellet can be summarized as follows: The Vardfjellet case is a graphite occurrence about 2.0 km × 0.5 km in size: it comprises at least eight individual subparallel and subvertical thick graphite rich lenses, which can be followed for at least 600 m along strike.

Our second example (Figure 8) is the occurrence Vikeid West (occurrence 16), which is almost completely covered with 1–2 m. thick peat. The presence of good conductors indicating graphite was first discovered by [98] and sampling from a selection of trenches was described by [108]. Figure 8a shows EM31 traversing superimposed on the airborne geophysics. High apparent conductivity from EM31 ground measurements coincides with low apparent resistivity from helicopter-borne measurements. Figure 8b shows the location of a short drill-hole, which penetrated several graphite lenses (see [60] for details). One of these lenses was used as a grounding point for charged potential (CP) measurements. The CP potential anomaly is shown with superimposed SP measurements points. The CP anomaly gives us quite accurate dimensions for this graphite lens and indicates a size of about 1200 m × 200 m. The electrical potential difference of <54 mV indicates a substantial downward extent. The thickness is interpreted to be the result of tightly isoclinally folded graphite lenses that have electric contact, and which are physically impossible to resolve by CP measurements. However, the continuous good conductivity measured by EM31 shows that we have a massive graphite mineralization with a considerable thickness. The SP data show, in addition, several graphite bodies adjacent to the CP anomaly, showing an SP anomaly exceeding 400 mV in several places. Based on EM31 data, eight exposures of graphite were discovered and sampled. Figure 8c shows the ERT/IP profile: its location is shown in Figure 8a,b. The data from the Vikeid west occurrences can be summarized as follows: From the airborne and ground geophysics, the occurrences have an extent to a depth of at least 100 m, they comprise at least 5 mineralized structures with an aggregate length of about 3500 m. The width of individual lenses varies from 10 to 100 m, the average being estimated to be ca. 40 m. Chemical analyses of 12 graphite schist samples from the area show an average % TC of 10.4%.

A similar approach was used for almost every occurrence on Figure 2a–c. This information gives us the number and apparent thickness of subparallel graphite lenses in the area and their indicated length along strike. The details for every occurrence can be found in [51,59,60,63].

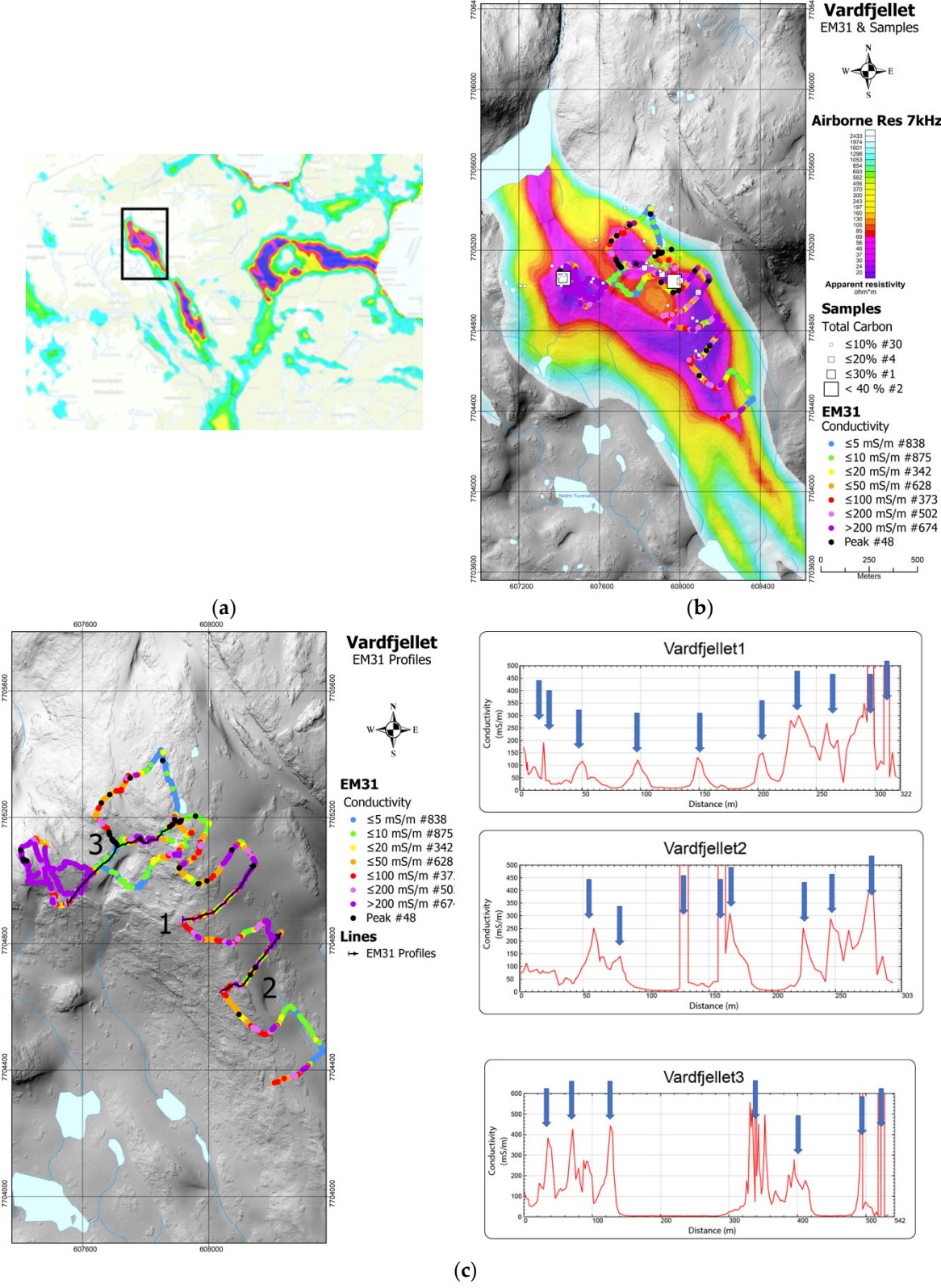

**Figure 7.** (**a**) Part of the airborne geophysical map showing the location of Figure (**b**). Airborne geophysical map with superimposed sampling points analysed for TC and measurements done with EM 31. (**c**) Sampling traverses of EM31 and, to the right, variation in apparent conductivity in three traverses, where peaks (blue arrows) show the location and apparent width of near-surface graphite lenses. This shows that airborne geophysics comprises several individual subparallel graphite lenses. A complexity that would not been resolved with airborne geophysics only. Data from occurrence two (Vardfjellet). Based on data from [51].

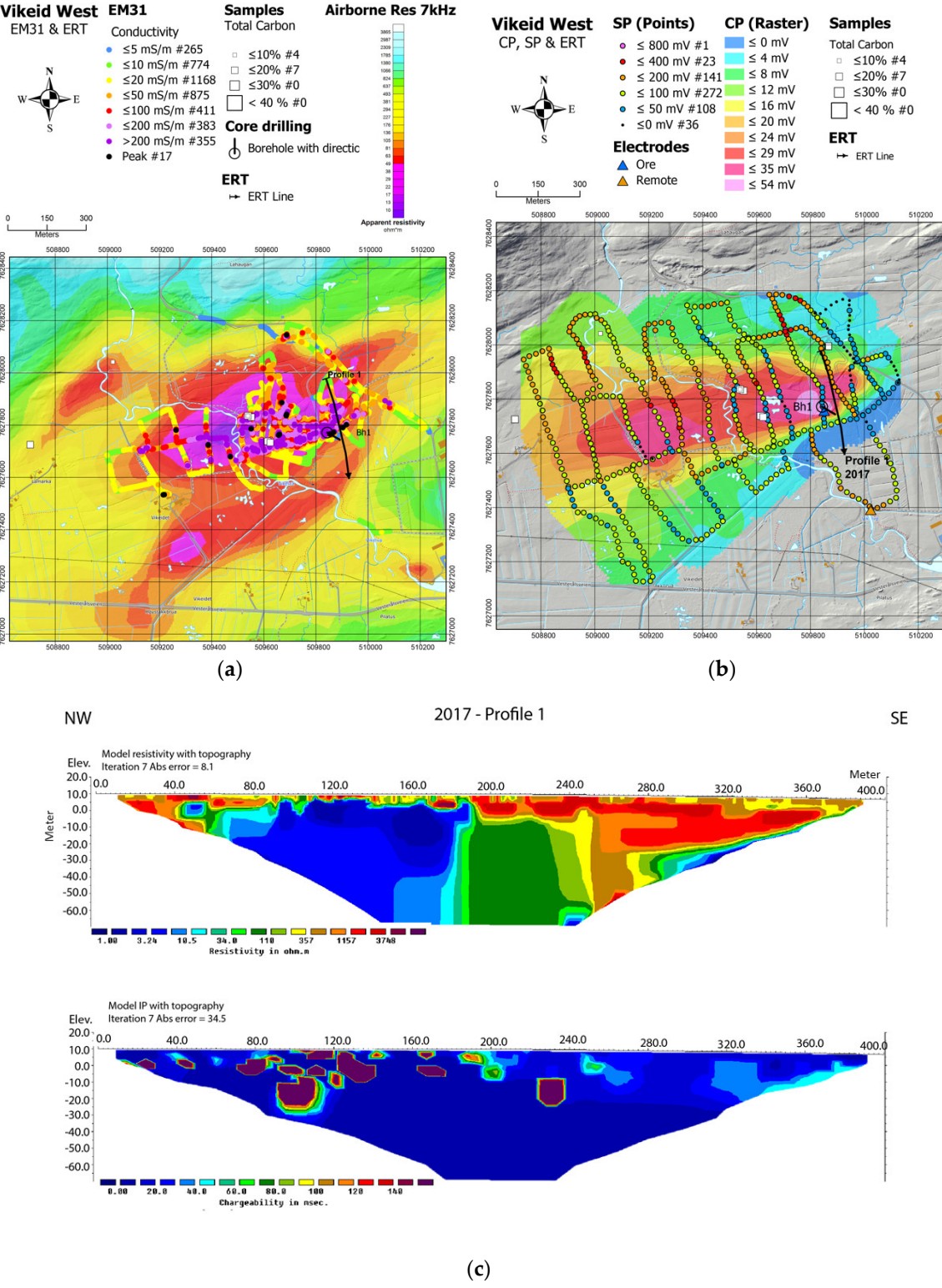

**Figure 8.** (**a**) Shows the airborne EM map with superimposed EM31 traverse, several highly conductive zones are apparent. (**b**) Grid map of CP measurements showing the dimensions of one individual graphite lens and superimposed measurements of SP, where several conductors are apparent. (**c**) ERT profile location shown on A and B. The upper part shows the variation in resistivity and soil cover is indicated. The lower part shows the variation in IP (induced polarization). Elevated IP shows the presence of electrically conducting material (graphite or sulphides). Based on data from [60].



*5.5. Resource Estimates*

Using the estimates of the length, the width, and the number of graphite lenses within the different occurrences, we need a common method to compare the inferred graphite resources. Despite the lack of drilling data from almost all of the occurrences, we still believe that a rough order of magnitude approach, in line with [119] could be used for the estimation of the volume of the respective occurrences. We have four main variables: *L* = length of mineralized zones in meters (from airborne and ground geophysics), *W* = average apparent width (m) of graphite zones (from EM31, drilling and observations), *α* = average dip in degrees from field observations (used to calculate real width), and *n* is the occurrence number in Table 1. If we also consider that the mineralizations are continuous 100 m down dip, the volume ($V_n$) of each occurrence would be:

$$V_n = L_n\ W_n\ (\sin(\alpha_n))\ 100 \tag{1}$$

Earlier reports used a density ($\rho$) of 2600 kg/m$^3$ [60,64]. In our calculation, we use $\rho$ equal to 2437 kg/m$^3$, which is the average from petrophysical measurements (see supplementary data). The tonnage will then be given, and if we regard the average % TC from Table 1 as valid. The amount of contained (Cg) graphite is calculated as follows:

$$Cg_n = V_n\ \rho\ (\%TC_n) \tag{2}$$

In each of the three provinces, we have the following averages: Senja (*n* = 6) average % TC and tonnage is 10.5% and 20.8 Mt, in Vesterålen (*n* = 18) 13% and 6.3 Mt, and in Holandsfjorden (*n* = 2) 7.6% and 1.25 Mt, respectively.)

The resource estimates calculated above (Table 1) are based on this approach and the results are sufficient for each occurrence to be defined as having an inferred resource with a low level of confidence.

*5.6. Beneficiation Results*

The beneficiations test done by [116] was recapped in [58] and can be summarized as follows: The head sample (300 kg) had a total carbon content of 17.4%. The head sample was crushed then ground in a ball mill with a pulp density of 67%. Grinding was optimized so that 90% is < 0.3 mm and 80% passing point was 0.15 mm. Starting with the head sample with 17.4% TC, after the different flotation and regrinding steps, the different concentrates had the following TC contents: rougher concentrate, 59.06% TC; cleaner concentrate 1, 79.48% TC; cleaner concentrate 2, 88.20% TC; and cleaner concentrate 3, 88.38% TC. The highest TC content, of 97%, was found in the >200-μm fraction of the cleaner concentrate 3 using Flotol B™ as a frother.

The trials reported in [52] were slightly different: the head sample (70 kg) had a TC of 19%. Following crushing, primary grinding was done in a rod mill with 45% pulp density and 12 minutes grinding time. Sieving with 100% <600 μm and de-sliming of the fraction <20 μm. Two different frothers were tested, Dowfroth 400E™ and Nashfroth 620™. Dowfroth 400E™ was found to be the best. Rougher flotation gave a rougher concentrate, with 75.9% TC. Two different regrinding runs, followed by two cleaning steps, gave a cleaner concentrate, with 98.13% TC and 97.13% TC, respectively [52]. Sieving of the cleaner concentrate gave for the three coarsest fractions a recovery of 72% and the following % TC in the size fractions: >300 μm 98.2%, <150> μm 98.1%, and <75> μm 90.4% (Table 2). In summary, the various cleaning steps gave a bulk concentrate with 90.1% TC and a recovery of 98.1%.

The results show that it is possible, with bench-scale beneficiation, to replicate results similar to what is achieved by the industry today using conventional methods. The coarser and more valuable flake fractions, in particular, have the highest purity. Given that the graphite schists seem to be of comparable mineralogy and metamorphic facies everywhere in the three graphite provinces, we believe that this shows, in all probability, that the graphite occurrences in Norway have a favorable mineralogy, suitable for beneficiation with no technical difficulties and using available technologies.

**Table 2.** Weight fraction (in % and cumulative), % TC recovered, % TC, and cumulative recovered, in cleaner concentrate (bottom row), from bench-scale beneficiation trials in [52]. In the size fractions > 150 μm the highest purity flakes (highest TC) are found.

| μm | Wt. % | Wt. % Cumulative | % TC | % TC Recovery | % TC Cumulative Recovery |
|---|---|---|---|---|---|
| 300 | 7.1 | 7.1 | 98.2 | 7.7 | 7.7 |
| 150 | 31.7 | 38.8 | 98.1 | 34.5 | 42.2 |
| 75 | 29.8 | 68.6 | 90.4 | 29.9 | 72.1 |
| 63 | 7.4 | 76.0 | 80.7 | 6.6 | 78.7 |
| 45 | 8.4 | 84.4 | 78.0 | 7.3 | 86.0 |
| <45 | 15.6 | 100.0 | 80.7 | 14.0 | 100.0 |
| Cleaner concentrate | 100.0 | | 90.1 | 98.1 | |

## 6. Discussion

### 6.1. Geology

The graphite-bearing rocks, in all the three graphite provinces, occur in what appear to be similar settings as can be summarized as follows: (1) the sedimentary protoliths to the graphite schists have an age in the range 2.2–1.7 Ga: this is based on their relative position to the country rock, since direct dating is difficult. (2) Petrography show similar mineralogy and metamorphic mineral assemblages both for graphite schists and for the different host rocks. Metamorphic conditions are generally upper amphibolite to granulite facies, in Vesterålen in the range 810–835 °C and 0.73–0.77 GPa [74].

The average deposit size (in Mt) clearly differs from Senja, with 20.8 Mt compared to Vesterålen where it is 6.3 Mt. A possible geological explanation for the differences could be related to the 1.8–1.79 Ga magmatic event [71]. In Vesterålen, where the AMCG rocks intruded and dismembered the older rocks, including the graphite units, to a much larger extent than on Senja. Younger intrusions are much less apparent on Senja. The degree of exposure of the graphite-bearing units is, however, small and the relationship and proportion between size of graphite occurrences and younger intrusions cannot be exactly quantified.

Most of the graphite deposits elsewhere on the Fennoscandian Shield occur in rocks of Paleoproterozoic age. The Nunasvarra graphite deposit, which is the largest graphite deposit in Sweden with its 12.3 Mt, 25.6% TC, occurs in carbonaceous sediments intermittent with tholeiitic volcanic rocks dated to 2144 +/− 5 Ma. The graphite is assumed to be formed in the subsequent Svecokarelian tectonometamorphic event [67]. Finland's largest graphite deposit, the Piipumäki deposit in eastern Finland, occurs in Svecokarelian graphite-bearing quartz-feldspar gneisses and amphibolites. Peak metamorphism is estimated to be 0.5 Gpa at 740 °C and dated at 1.8–1.79 Ga [57]. Graphite deposits across the Fennoscandian Shield have similar geological settings regarding lithology, age of host rocks, and peak metamorphic conditions. A possible hypothesis is that the high metamorphic-grade flake graphite occurrences in Norway, Sweden, and Finland originally represented rocks that were formed in the same type of setting as, and contemporaneous with the low-metamorphic grade shungite rocks of Russian Karelia on the easternmost perimeter of the Fennoscandian shield. The 2.0 Ga old upper part of the Zaonezhskaya Formation near lake Onega comprises a 600 m thick succession of bitumen-rich siltstones and tuffs. In these low greenschist facies rocks TOC are, on average, 25% but can reach levels up to 98%. The shungite rocks comprises black, dense amorphous and nanocrystalline organic matter. Shungite rocks also represent the world's biggest accumulation of organic matter in the Paleoproterozoic, believed to have been formed in a brackish water environment [120,121]. In Norway, the multistage high-grade metamorphism combined with polyphase deformation have removed almost all signs of primary features in the rocks associated with the graphite occurrences. The ages of the Norwegian graphite occurrences and the shungite rocks seems to coincide. Isotope data could give some circumstantial evidence for a similar origin. However, it would also be difficult to find conclusive evidence due to the large difference in the tectono-metamorphic history of these deposits.

### 6.2. Comparison of Resource Estimates

For evaluating graphite deposits from an economical point of view [122], six key factors can be used to rank the economic potential among deposits:

(1) % TC (grade), deposit size, and contained graphite, were % TC rank before deposit size.
(2) Location (country risk).
(3) Flake size distribution.
(4) Product purity.
(5) Off-take agreements.
(6) Time frame of production.

The two latter are not relevant for the stage of development of our occurrences. In Table 1, a number of the occurrences would have tonnages that many would regard as (too) small. To test this, we compiled resource data from graphite deposits and junior companies that are noted on the Sydney (ASX), Toronto (TSX), and London (LSE) stock exchanges (see supplement data for details). In Figure 9, grade, tonnage, and contained graphite are compared. Except for some very large international deposits, the levels and variation for the north Norwegian graphite occurrences are in the same of order of magnitude as those found elsewhere. We believe that this shows that our rough resource estimates are realistic and that the graphite provinces in Norway have the same prospectivity as those in other parts of the world. For factor two, the Norwegian occurrences would rank among the top in the world with respect to their geographical location near to harbors and from a political governance point of view, as measured by Worldwide Governance Indicators [123]. For factor three, flake size distribution, both in thin section, and most importantly in a crushed product, are also favorable. Lastly, the product purity, factor four, is comparable to what is achieved from the graphite industry, with the use of flotation alone.

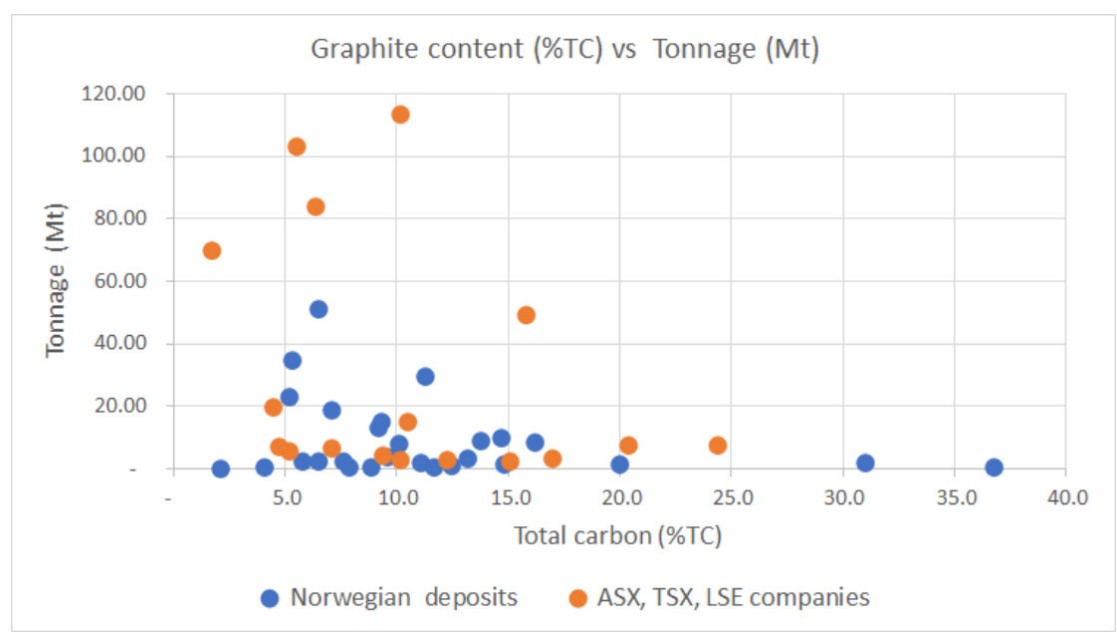

(**A**)

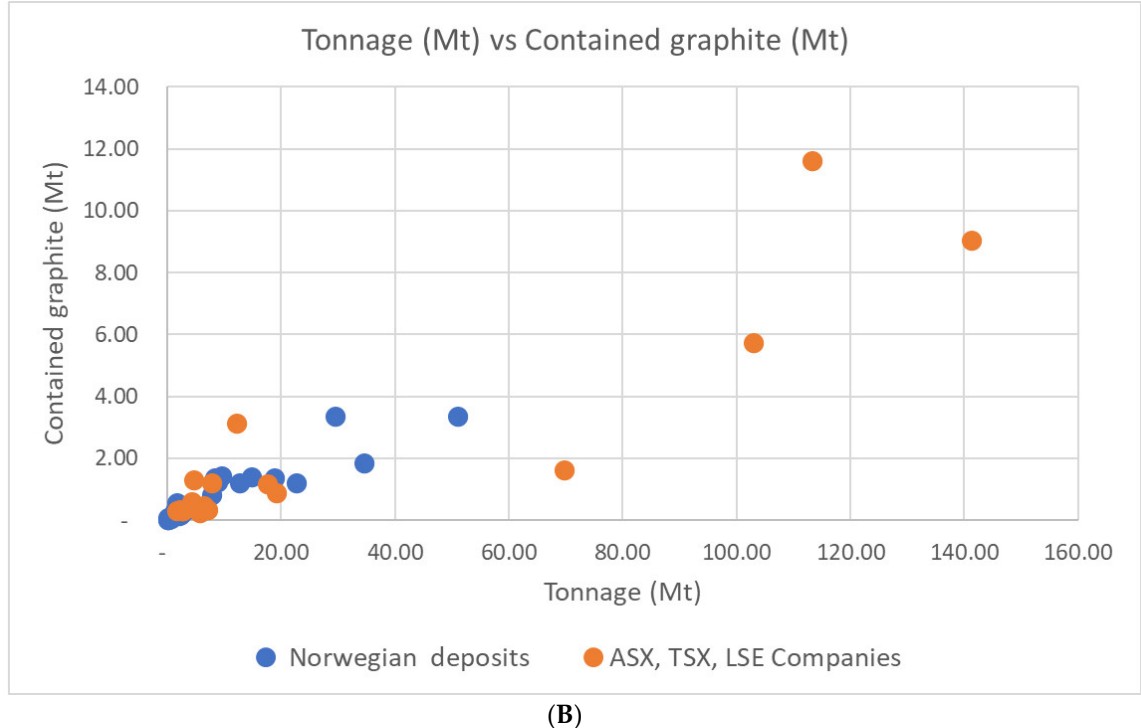

(**B**)

**Figure 9.** Comparison of graphite content (% TC), tonnage (**A**) (Mt), and contained graphite between (**B**) North Norwegian graphite occurrences and deposit data from companies listed on ASX, TSX, and LSE stock exchanges (see supplementary data).

## 7. Summary and Conclusions

Graphite is abundant at a number of localities in three graphite provinces in Northern Norway. The provinces show similarities in geological setting with regards to: (a) host rocks for the graphite mineralization; (b) metamorphic history of the graphite host rock; and (c) age of the associated country rocks, which pre and postdate the graphite-bearing succession. The graphite-bearing rocks in all the provinces appear to be younger than 2.7–2.6 Ga Archean gneisses and older than granitic to charnockitic intrusions of 1.8 to 1.7 Ga. We speculate that the Norwegian graphite occurrences may represent equivalents to the shungite rocks of Russian Karelia.

Ground geophysical measurements show that the many of the occurrences have a complex internal structure, commonly comprising several individual graphite lenses that have undergone polyphase deformation, which will require detailed structural mapping. The graphite schist has a composition similar to what would be classified as meta-pelites to meta-arenites: traces of primary sedimentary structures are rarely observed. Our resource estimations of the graphite resources give an average of 9.3 Mt with 11.6% TC and 0.81 Mt of contained graphite for all the 28 reported occurrences. The aggregated graphite resources are 241 Mt of graphite ore with 21.5 Mt of contained graphite. The size and graphite content are in the same range as what is reported by a number of graphite companies elsewhere in the world.

Bench-scale beneficiation trials show that is possible to achieve purity of graphite concentrate of above 98% TC in size fractions >300 µm. It is concluded that there are no technical difficulties in producing products of good quality. In summary, the north Norwegian graphite provinces are highly prospective from an economic point of view. They also include several occurrences where graphite that has been formed deep in the crust can be studied in detail.

**Supplementary Materials:** The following are available online at www.mdpi.com/2075-163X/10/7/626/s1.

**Author Contributions:** All authors contributed to this manuscript. Conceptualization of manuscript: H.G.; Conceptualization and administration of over-arching projects: J.S.R. and H.G.; Writing of manuscript: H.G.,

J.S.R., A.K.E. and I.H.C.H.; Data collection and processing: B.E.L., J.G., H.G., J.S.R., J.K.S., A.K.E., I.H., F.O., H.E. and B.D. All authors have read and agreed to the published version of the manuscript.

**Funding:** This research was funded by Nordland and Troms counties, Geological Survey of Norway.

**Acknowledgments:** A number of people have contributed to the graphite exploration in Norway. Particularly we thank Jan Egil Wanvik and Nils Egil Johannessen. We also thank Rasmus Blomquist and Audun Sletten for the release of data from Norwegian Graphite to NGU. Over the years a number of local people contributed with logistic field support, we thank particularly Ola Grindvold†, Bjørn Larsen†, Johan Jakobsen, and Per Mattis Skum. Rognvald Boyd, Hanne-K. Paulsen, and the reviewers are thanked for comments to the final version of the text. Due to the uncertainty that naturally is associated with interpretation of geological and geophysical data, neither the authors nor the Geological Survey of Norway accept any liability for any actions third parties may take when using the information in this review or from the references.

**Conflicts of Interest:** The authors declare no conflict of interest.

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
