# Peer review of "The Graphite Occurrences of Northern Norway, a Review of Geology, Geophysics, and Resources"

_minerals, doi:10.3390/min10070626_

Round 1

Reviewer 1 Report

Review of
“The graphite occurrences of Northern Norway, a review of geology, geophysics and resources.”

By

Gautneb et al

The introduction needs to state that this report will not provide predictions to where new graphite deposits may be found, but reviews the existing known occurrences. I am aware that the title does state ‘review’, so the framing of the paper is implied, but nonetheless this statement should be made to frame the scope of the paper explicitly.

A short one paragraph history of the graphite commodity price, say over the same period that exploration/mining has been conducted would be a useful addition to put the value of graphite in context.

I appreciate the efforts of the authors to synthesize the geology of the various deposits into a generic description. Listing the commonalities is a useful part of this paper, and helps the reader understand where to look for graphite potential. One thing that is missing is an example where all the different data sources are presented. There are good examples outcrops, drilling, microstructure, mapping, geophysics but they all from different deposits (figs 3 to 8). I think one of these deposits could be selected to produce a figure that shows at least a few of these properties simultaneously. A suggestion is occurrence 25 where there is drilling. Can the surface mapping be shown with the collar locations? A geophysical image showing the EM response with the map would be very useful. A plot of downhole resistivity log with the drill litho logs would also help he reader understand the relationships between lithology, graphite and resistivity – a very important aspect for exploration. This would then help interpretation of the surface geology and EM geophysics.

The ‘Resource estimates’ section and the ‘beneficiation of of graphite schist’ sections belong in the methods. The final paragraph (lines 411 to 416 and table 2) can remain in the results.

Resource estimates: Given the lack of data, establishing a solid link between resistivity and the presence of graphite (see previous comments re plotting downhole resistivity logs with litho logs) is very important. This needs to be done. In addition, given density is selected as a variable for the resource estimate (line 382), establishing how density predicts graphite is also needed, but is currently missing. A section describing the petrophysical characterisation of graphite deposits is needed, with a table/figure displaying this clearly.

The manuscript is structured well (some changes listed above need addressing) however the grammar requires some work. In particular, use of plurals needs correcting. I assume the authors will redraft the manuscript during revision. Additional attention to writing style and grammar would benefit the quality of the manuscript.

Author Response

Dear reviewer 1

I thank the reviewer for your constructive comments, many of the comments are addressed also in the covering letter to the editor, please consult this also

When referring to line numbers below they are according to the present version of the text

We thank for helpful comments.

.

Below is my comments

A sentence that we regard ore predictive modelleing is added

On line 36-40 we added a paragraph on graphite price range, However its is important to point out that the increased interest in graphite that we have seen in the last years it not related to increased price or shortage, but to criticality particularly the concentration of worlds production in a small number of countries particularly China and a potential supply risk related to the geopolitical situation.

  1. The section no 3 in reviewers comment is very difficult to comply, as written in the covering letter there is no single locality that can be used to show all the aspects of the exploration work simultaneously. Locality no 25 the Jennestad is actually a very bad place to get an impressions on surface geology of the graphite bearing rocks, ground geophysical mapping has not been done at this locality and we have used Norwegian minings own resource estimate in Table 1. The area is also for a large degree overburden, any good geological mapping including structural studies can not be done there. Norwegian mining put their drill holes in this area because this was the place were they had the mining agreement with the land owner and because this locality is not far from some old graphite mines.

What the locality with the drilling represents is, the only place several where +100 m continuous sections showing the distribution graphite rich layers in relation to their immediate country rock. No other place is this seen.

Reviewer 1 also want to see down hole resistivity plots. Unfortunately Norwegian mining did not do any geophysical drill hole logging, such data do not exist.

I must say that I was quite in despair when read this requirement from reviewer 1. If the reviewer and editor insist on resistivity data to be included for the drill logs, the whole paper has to be rejected or withdrawn. The lack of these data totally beyond our control. I have addressed this issue particularly in the covering letter to the editor and asked for acceptance of the logs without the resistivity.

Reviewer 1 apperantly wants the most of section on resource estimates and beneficiation test moved to chapter of methods. A methodical introduction to the benefication work is in the methods chapter already . However since the resource estimate chapter is based on and is justified by the methods described in the previous chapter on Ground geophysical measurements it seem unnatural to present the use of the geophysical data earlier in the text than the description of the data that the resource estimates is based on.

Reviewer both reviewer 2 and 3 seem to approve the present order of chapters which follows what is normal in research papers.

I ask for the editors decision regarding the sequence of the sub chapters to be accepted

4) Resource estimates: here again the Reviewer 1 explicitely requires resistivity measurements to be plotted down the log together with litho data. Again this can not be done since such data never have been collected, by does who gave us the log data. and had the data been available they would on be relevant for 1 of 28 occurrences.

We agree with Reviewer 1 that petrophysical data is useful and have included in the supplement data an excel sheet that includes 126 samples were volume, density, pore volume porosity, susceptibility, remanens heath conductivity, specific heath capacity and lastly total Sulphur and carbon are measured. The graphite schist has an average density of 2437 kg/m3. We used this density to correct the resource estimates and thus all values corrected with a factor of 0.94.

Reviewer 2 Report

This paper presents an overview of potential economic graphite occurrences in northern Norway. The paper is largely based on existing, mostly Norwegian data. The paper demonstrates that there is potential for graphite mining in Norway. To my knowledge, this is the first paper that describes graphite occurrences in Norway.

The paper is clear and to the point. The English needs to be checked carefully though. I have made some language corrections in the annotated PDF file.

I have the following general comments:

(1) The graphite occurrences are situated in high-grade metamorphic rocks. From the descriptions and the photographs, it appears that most of the graphite is the result of the transformation of organic in the sedimentary protoliths during prograde metamorphism. It would be good if the authors could confirm (if correct) this in the text.

(2) Is there any evidence for hydrothermal graphite? One would expect that during prograde metamorphism the H2O released from dehydration reactions reacts with the organic material to form a carbon-bearing (CO2 and/or CH4) fluid. The retrograde metamorphism, this fluid would precipitate graphite.

(3) Carbon stable isotopes could be used to link the Norwegian graphite with the graphite in Shungite rocks.

(4) The authors could include a section on future work, including XRD, Raman spectroscopy, carbon stable isotope, and fluid inclusion work.

Detailed comments:

Fig. 6: Scale is missing

Fig. 6: It is not cleat what exactly the dashed lines for F2 and F3 folds indicate. Do these lines represent the traces of the fold axial planes?

Fig. 7 and 8: Text font size in the legend is too small.

More minor corrections/commnes are given in the annotated PDF file.

Author Response

Dear reviewer no 2

I thank the reviewer for his constructive comments, many of the comments are addressed also in the covering letter to the editor, please consult this also

When referring to line numbers below they are according pdf annotated by the reviewer 2 and differ from the submitted ms.

First I thank the reviewer 2 for his careful correction of many grammatical and linguistic errors. They are all taken into account.

Reviewer 2 is correct when he states that this is the first paper that describes all the graphite deposits of northern Norway if accepted.

  • We agree with the reviewer 2. The geological setting of the graphite horizons, in all probability shows that they were formed in a sedimentary succession and where converted to graphite during metamorphism. We added a sentence at line 256 to clarify this.
  • this is a good question we have found some few localities where graphite could be interpreted as being remobilized, they represent small curiosities and have not been studied in detail (yet). In southern Norway Bamble area there are very nice examples of vein type graphite, work with them are in progress.
  • We have some isotope data and the work to compile this is in preparation. Since this paper have its aim to review the exploration work, we feel that to include isotopes is beyond the scope of this paper. The paper is already quite comprehensive. I request that the reviewer accept this.
  • As said in point 3) we are working on this, Raman has been documented in two other studies reference 25 and 56. Since the peak metamorphic temperature has been in the order of 810-835 well above peak for the Raman based thermometer, it is not clear to me to what degree Raman will add new data. In additional to what is published

Detailed comments

Fig. 6. Scale. The scale on fig. 6 is given on the outer grid, however a scale is added for clarity.

Fig 7 and 8 size of legend text increased

Line 65 updated and clarified

Line 70 sentence updated and clarified

Line 302-306 figure text updated to include info on plunge.

Line 391 Table 1 is updated to clarify that n = the occurrence number.

Line 476 sentence re written to specify that the age of the sedimentary protolith is assumed to be 2.2-1.7 Ga

Line 491 age corrected.

Line 501 the requested information related to the Zaonezhskaya Formation is added.

Line 494 to 496 re-written.

Line 545-547 re-written for clarity.

Reviewer 3 Report

Review of the MS

“The graphite occurrences of Northern Norway, a review of geology, geophysics and resources”

by

Håvard Gautneb, Jan Steinar Rønning,  Ane K. Engvik, Iain H.C. Henderson, Bjørn Eskil Larsen, Janja Knežević Solberg, Frode Ofstad,, Jomar Gellein, Harald Elvebakk, and Børre Davidsen

The submitted MS is basically a slightly extended report on exploration survey which from a scientific point of view does not bring any brand new information that would be of interest to the general professional community. On the other hand, it is a very carefully prepared manuscript which is easy to read and provides a comprehensive overview of the knowledge about Norwegian graphite deposits and particularly about the current methods of their exploration. From this point of view, I especially appreciate the part of the work that deals with the use of airborne geophysical methods and verification of their results by the ground survey. The part of the work devoted to the calculation of graphite reserves  and to processing of graphite ores is done in a standard way. The work is written clearly, in good English.

I have a few comments on the submitted paper:

Introduction

Lines 36 až 39: …Somewhat simplified, graphite mineralization is the end result of essentially two processes: The graphitization of carbonaceous material (produced from photosynthesis and other biogenic processes) by metamorphism, or the deposition of graphite by hydrothermal processes with different carbonaceous sources….

General comment: This statement is really simplified because the original organic matter in the processes of epigenesis and low-grade metamorphism (up to and including the greenschists facies), is actually gradually carbonized and during this carbonization, the heteroatoms are released as volatiles and remaining solid phase is step by step enriched with carbon. Original turbostratic structure of organic matter is gradually replaced by small „stacks“ of homogeneously oriented domains, called Basic Structural Units (BSU; Oberlin and Bonnamy, 2001). The BSU gradually coalesce to isochromatic domains (isochromatic lamellae). However, this process does not result in the formation of graphite, which originates by decomposition of the initial organic matter into a mixture of gases (mainly CO2 and CH4) and crystallizes from this mixture under amphibolite or granulite facies conditions according to the equation: CO2 + CH4 → 2C + 2 H2O (for review see, for example, Buseck and Beyssac, 2014, Elements , vol. 10, pp. 421-426). If this process takes place in situ, then the disseminated crystalline flake graphite deposits are formed, as exemplified by the Norwegian deposits. When CO2 and CH4 migrate in hydrothermal fluids, then the vein-type lump or chip graphite deposits occur in high-grade metamorphic regions (graphite deposits in Sri Lanka).

The chapter “Geological Setting” This chapter is very brief, with a number of relevant references.

The chapter "Previous and recent work" is also clearly arranged.       

The chapter “Materials and methods” This chapter summarizes the geophysical, geological and geochemical methods  used in the exploration of graphite deposits,  and provides basic data on the mineral processing technology of graphite ores. I have no comments on this chapter.

The chapters “Results”, sub-chapters “Geology of the graphite bearing units”, “Petrography of graphite schist” and “Structural geology of the graphite bearing units”: Comment: These paragraphs of the paper are very descriptive so that they are not much interesting.

The sub-chapter “Ground geophysical measurements”. Comment: the title of this chapter should be modified as both the results of the ground and  airborne geophysical surveys are given here. I consider this sub-chapter as the best one of the entire paper.

The sub-chapter “Resource estimates”. Standard procedures were used to calculate the graphite ore reserves.

The sub-chapter “Beneficiation of graphite schist”: Comment: In this chapter I am missing the data on the recovery of flotation concentrates in wt % related to the original weight of the graphite ore (the yield of flotation concentrates in individual flotation steps in wt % of the head sample). If the recovery is low, then the flotation process appears to be economically inefficient.

The sub-chapter “Geology”: I consider the comparison of Norwegian graphite deposits with those of shungite  in Karelia to be adequate. It is suitable to remind that some of the shungites were metamorphosed to form graphite.

The sub-chapter “Comparison of resource estimates”.  I have no comments on this sub-chapter

Lines: 496-498: The graphite schist has a composition similar to what would be classified as meta-arenites, but traces of primary sedimentary structures are very rare to observe. Comment: This statement is not justified in the text. High contents of organic carbon are mostly associated with pelitic sediments, and very often with a tuffitic admixture. Their metamorphosis produces amphibole-biotite gneisses.

I do recommend the paper to be accepted for publication after slight modifications and a few complements.

Author Response

Dear reviewer 3

I thank the reviewer for his constructive comments, many of the comments are addressed also in the covering letter to the editor, please consult this also

When referring to line numbers below they are according to the present version of the text

Introduction the paragraph on graphite formation, I thank the reviewer for correcting this over simplification, I have tried to re- write the text accordingly.

Geological setting no other changes except for linguistic corrections

Previous and recent work:, no changes, except grammar. I appreciate the acceptance of this historical account, this gives us an opportunity to refer to works that is not commonly known.

Materials and methods: this chapter only has changes based on comments from the other reviewers.

Results; sub chapters sub-chapters “Geology of the graphite bearing units”, “Petrography of graphite schist” and “Structural geology of the graphite bearing units”. I agree that these chapters does not have the highest scientific level, however I believe that they are important they give examples of extraordinary well exposed localities of graphite schist.

Results; sub-chapter on Ground geophysical measurements. I agree that this chapter contains the core results of our paper, to use ground geophysics to demonstrate internal complexities in airborne geophysics is not common to do .

Sub chapter on Resource estimates. Here I feel that reviewer 3 oversimplifies when he states that standard methods is used. The standard methods, as I see it, would be to follow the international mining standard like the Australian, JORC or Canadian National Instruments 43-101 standard. We have not done that, but used as I see it the data that we have. In addition what we have done in the discussion chapter is to compare our results with resource estimates from stock exchange traded mining companies (that follow the mining standards) and show that our results are similar, and thus by all probability realistic. I believe that this give important information to any new mining company that want to explore fore graphite in Norway. Reviewers comments on this is not a crucial point in the paper. however I hope I have addressed this appropriately.

Beneficiation of graphite schists. I renamed the chapter to Beneficiation results and included a sentence to summarize these results. The beneficiations trials can in summary state: the various cleaning steps gives a bulk concentrate with 90.1% TC and a recovery of 98.1%. A good point by the reviewer, that I had forgotten, which now is addresses

Discussion on geology: All reviewers seem to find the comparison between the Svcandinavian graphite and shungites as adequate.

Line 559-561 its is most correct to say that the graphite schist has a meta pellitic to meta arenitic composition, I agree with the reviewers comments here, and have changed accordingly.

Many thanks

Håvard Gautneb
